# Can We Achieve More with Less? Parenchymal Sparing Surgery Versus Major Liver Resection for Colorectal Liver Metastases: An Observational Single-Center Study with Propensity Score Analysis

**DOI:** 10.3390/diagnostics15111334

**Published:** 2025-05-26

**Authors:** Sorinel Lunca, Stefan Morarasu, Raluca Zaharia, Andreea-Antonina Ivanov, Cillian Clancy, Luke O’Brien, Wee Liam Ong, Gabriel-Mihail Dimofte

**Affiliations:** 1Department of Surgery, Grigore T. Popa University of Medicine and Pharmacy, 700115 Iași, Romania; sdlunca@yahoo.com (S.L.); raluca.zaharia11@yahoo.com (R.Z.); i_antonina@yahoo.com (A.-A.I.); william05021990@gmail.com (W.L.O.); gdimofte@gmail.com (G.-M.D.); 22nd Department of Surgical Oncology, Regional Institute of Oncology (IRO), 700483 Iasi, Romania; 3Department of Colorectal Surgery, Tallaght University Hospital, D24 YN77 Dublin, Ireland; clancyci@tcd.ie (C.C.); lukemobrien13@gmail.com (L.O.); 4Trinity College, University of Dublin, D02 PN40 Dublin, Ireland

**Keywords:** liver surgery, parenchymal sparing, hepatectomy, major liver resection, morbidity, overall survival, disease-free survival

## Abstract

**Background/Objectives:** Colorectal liver metastases (CRLMs) occur in 25–30% of colorectal cancer (CRC) patients, significantly impacting survival. While major liver resection (MLR) was traditionally preferred for oncologic clearance, parenchymal-sparing surgery (PSS) has emerged as a less invasive alternative. This study compares perioperative and long-term outcomes of PSS versus MLR in CRLM patients. **Methods**: We conducted a retrospective cohort study at the Regional Oncology Institute, Iasi, Romania, analyzing patients who underwent hepatic resection for CRLM between August 2013 and June 2024. Patients were categorized into PSS (n = 58) and MLR (n = 28) groups. Outcomes assessed included perioperative parameters, postoperative morbidity, overall survival (OS), and disease-free survival (DFS). **Results**: PSS was associated with a shorter operative time (235.2 vs. 302.6 min, *p* = 0.003), lower morbidity (18.9% vs. 57.1%, *p* = 0.001), and fewer major complications (Clavien–Dindo ≥ III, *p* = 0.005). ICU stay was significantly longer in MLR patients (*p* = 0.04). After propensity score matching (PSM), PSS was found to have lower morbidity compared to MLR (*p* = 0.023) with similar major morbidity (*p* = 0.473) and LOS (*p* = 0.579). Overall survival (31 vs. 37.1 months, *p* = 0.884) and disease-free survival (25.2 vs. 22.2 months, *p* = 0.519) were comparable between the groups before and after propensity score matching PSM (40.9 vs. 21.2 months, *p* = 0.741 and 24.3 vs. 13.8 months, *p* = 0.653). **Conclusions**: PSS achieves comparable oncologic outcomes to MLR while reducing postoperative morbidity and ICU stay. These findings support PSS as the preferred approach for CRLM, reserving MLR for select cases requiring extensive resection.

## 1. Introduction

Colorectal liver metastases (CRLMs) are found in 25–30% of colorectal cancer (CRC) patients and are the second leading cause of cancer-related death in the Western world [1]. The 5-year survival rate for CRLM has now surpassed 50%, thanks to advancements in multidisciplinary management [2,3]. This progress is driven by a combination of surgical resection, multi-modal chemotherapy, and various ablation techniques. Among these treatments, liver resection remains the cornerstone of therapy, as 5-year survival is highly unlikely without surgery [2,3,4]. Achieving negative microscopic margins while ensuring an adequate functional liver remnant is key to successful surgical outcomes [5].

Major liver resections (MLRs) were traditionally performed to achieve an R0 resection. However, these extensive resections often came with high morbidity, particularly liver failure, and increased mortality rates [6,7]. Recent studies have shown that a margin of at least 1 mm is sufficient to achieve survival outcomes comparable to those of wider margin resections [5,8,9]. Moreover, with the advancements in modern chemotherapy, R1 resections by necessity can provide survival outcomes that are not significantly lower than those of R0 resections [10,11].

The gradual expansion of liver resection criteria, particularly the reduction of resection margins and the emphasis on preserving functional liver parenchyma (salvageability), has given rise to the concept of parenchymal-sparing surgery (PSS). While there is still debate in the literature regarding the optimal resection approach for colorectal liver metastases, an increasing number of studies support the use of PSS. Research suggests that PSS can provide oncological outcomes comparable to MLR in terms of overall survival (OS) and disease-free survival (DFS), while also reducing morbidity and mortality [12,13,14,15]. Despite the abundance of studies investigating surgical strategies for colorectal cancer liver metastases (CRLMs), the overall quality of evidence remains suboptimal, necessitating caution when interpreting findings. Many studies are retrospective in nature and exhibit heterogeneity in the surgical techniques applied, ranging from PSS to MLR, which introduces potential selection bias. Key prognostic variables—such as the number, size, and anatomical location of metastases, along with characteristics of the primary tumor including site, molecular profile, and histological grade—were often inadequately reported. Furthermore, critical treatment factors, particularly the rationale for neoadjuvant or adjuvant chemotherapy, were frequently omitted, despite their recognized importance in current clinical guidelines [15,16,17]. These factors may influence the results of individual studies and change the subsequent perception favoring PSS in terms of oncological safety.

Considering the preexisting debate regarding the oncological safeness of PSS, we aimed to retrospectively compare the perioperative and long-term outcomes of PSS and MLR to determine if PSS is associated with lower postoperative morbidity and mortality, without compromising cancer-related outcomes.

## 2. Materials and Methods

### 2.1. Design and Setting

A single-center retrospective observational cohort study was conducted at a tertiary cancer center—Regional Oncology Institute (IRO) Iasi, Romania—on patients with colorectal liver metastases who underwent hepatic surgery from August 2013 to June 2024. All surgical procedures were performed at the Regional Oncology Institute (IRO) Iasi, Romania, by a single surgeon from the 2nd Department of Surgical Oncology. Our study was designed retrospectively, making randomization impossible. A standardized follow-up protocol was established for all patients. From the time of hospital discharge, patients were monitored through outpatient visits every three months during the first year. From the second to the fifth year, follow-up intervals were extended to every six months. Each follow-up included routine blood tests, tumor markers such as carcinoembryonic antigen (CEA) and carbohydrate antigen 19-9 (CA19-9), abdominal MRI every six months, and chest CT scans. Positron emission tomography (PET) scans were performed when there was clinical or radiological suspicion of recurrence or distant metastasis. Patient consent for this specific study was waived by our institutional ethics committee, as the research posed no risk to the patients, and all collected data were anonymized. However, all patients admitted in our department, if they agree, sign a general consent form to use their medical data for research purposes.

### 2.2. Inclusion and Exclusion Criteria

The inclusion criteria consisted of patients with CRLM undergoing a first liver resection without extra-hepatic disease. Exclusion criteria included patients with a history of previous liver resections and those with multiple metastases requiring both parenchymal-sparing surgery and major liver resection. (Figure 1).

### 2.3. Data Extraction

Data were obtained from a prospectively maintained electronic database on patients with CRLM, including perioperative, short-term, and long-term outcomes data. Preoperative evaluation included information about demography (age, sex, BMI, smoking, and alcohol status), diagnosis (primary colorectal tumor location), comorbidities (hepatic, cardiac, respiratory, metabolic, or renal disease), and neoadjuvant therapy. Characteristics of metastases and the liver surgery included number and size of CRLM, type of liver resection, involved Couinaud segments, ASA score, intraoperative blood loss, number of postoperative complications, staged surgeries, operation date, duration of surgery, presence of extrahepatic metastases, and histopathological parameters (histological type of cancer, tumor differentiation, lymphovascular invasion, perineural invasion, resection margins). The recorded postoperative data included complications classified according to the Clavien–Dindo system, with major complications defined as Clavien–Dindo ≥ III, as well as reoperation rates, intensive care unit length (ICU) of stay, 30-day mortality, overall survival, and disease-free survival. Recurrence was diagnosed based on clinical follow-up, which included computed tomography scan, magnetic resonance imaging, and PET scans of the chest, abdomen, and pelvis, following the protocols established at our center.

### 2.4. Statistical Analysis

Continuous variables were presented as mean ± standard deviation (SD), median, and the values of quartiles Q25 and Q75. The reference threshold for the significance level *p*-value was 0.05. All data analysis was performed via XLSTAT v2021.3.1. (Lumivero^®^) software. The Fisher exact test for qualitative variables was used instead of the Chi-square test when the number of datasets and theoretical counts was less than five. The *t*-test (pooled variance) was used for quantitative variables to compare means. The normality of the quantitative variables was assessed using the Shapiro–Wilk test prior to applying the *t*-test. A *p*-value < 0.05 (95% CI) was considered significant. Kaplan–Meier survival curves were performed to compare DFS and OS between the two groups. For survival analysis, median survival time, rather than mean, was used as the median survival time is insensitive to outliers. The Cox proportional hazards model was used to assess whether PSS increases the risk of recurrence or death over the follow-up time. To further reduce bias from confounding variables and selection bias, propensity score analysis (PSM) was performed using XLSTAT software as previously conducted [18,19,20]. Matching control patients in the MLR group were selected according to propensity scores, in a 1:1 ratio, determined by Mahalanobis distances, with patients in the PSS group, based on covariates including age, gender, BMI, history of cirrhosis, smoking, diabetes, congestive heart failure, chronic obstructive pulmonary disease, chronic kidney disease, ASA score, and TNM staging (for survival curves).

## 3. Results

### 3.1. Patient Characteristics

A total of 86 patients were included in the study, split into two groups: parenchymal sparing surgery (n = 58) and major liver resections (n = 28). Mean age was 60.7 vs. 59.8 years old. There were significantly more male patients in the PSS group (39 vs. 12, *p* = 0.03). BMI was similar between the two groups (25.2 vs. 26, *p* = 0.343). The two groups were similar in terms of preoperative comorbidities, including history of alcoholic liver disease (ALD), cirrhosis, diabetes, chronic heart failure (CHF), chronic obstructive pulmonary disease (COPD), and chronic kidney disease (CKD). History of smoking, neoadjuvant therapy, and American Society of Anaesthesiologists (ASA) scores were similar. Regarding operative variables, the operative time was significantly longer for MLR (235.2 vs. 302.6 min, *p* = 0.003); however, intraoperative blood loss was similar (567.6 vs. 632.6 mL, *p* = 0.541) (Table 1).

### 3.2. Postoperative Outcomes

Overall, there was a higher rate of morbidity in the MLR group (*p* = 0.001) and a significantly higher rate of major complications (Clavien–Dindo III–IV) (*p* = 0.005), although when analyzed separately, the two groups were similar in terms of rate of postoperative bile leaks, posthepatectomy liver failure (PHLF), surgical site infections (SSI), and medical complications. MLR was associated with a higher rate of reintervention (*p* = 0.03) and a longer ICU stay (*p* = 0.04). After PSM, the rate of morbidity remained more frequent in the MLR group (*p* = 0.023). Most complications showed a higher frequency in the MLR group; however, they did not reach statistical significance. Length of ICU stay was longer in the MLR with a mean difference of 2.4 ± 5.5 days, without reaching significance (Table 2).

### 3.3. Long-Term Outcomes

With a mean follow-up of 33 months, patients who underwent PSS had a mean OS of 31.0 (3–110) months compared to 37.1 (3–123) months for those who underwent MLR (*p* = 0.341). Similarly, the mean DFS was 25.2 (4–96) months for the PSS group versus 22.2 (4–120) months for the MLR group (*p* = 0.605). (Table 3). In a univariate Cox proportional hazards model, PSS was not found to be a risk factor for death (OS) or recurrence (DFS), with a hazard ratio of 1.133 [95% CI 0.565–2.273, *p* = 0.724] and 0.897 [95% CI 0.537–1.497, *p* = 0.677], respectively. After, PSM results remained similar both for DFS and OS. Kaplan–Meier curves (Figure 2 and Figure 3) depict the survival distribution between the two groups showing similar OS and DFS survival.

In a multivariate Cox proportional hazards model for OS and DFS including PSS, positive resection margin rate, size of lesion (greatest diameter more than 50 mm, to be classified as large), and number of lesions (more than 2) were the covariates to influence OS (Size, Wald Chi^2^ 20.727, *p* < 0.0001, HR 6.895 [3.003–15.832]) and DFS (Number, Wald Chi^2^ 7.763, *p* = 0.005, HR 2.451 [1.305–4.606]) the most. From this model, the use of neoadjuvant therapy was excluded due to the low number of non-events (patients not having neoadjuvant therapy). The designed model, formed by the above four covariates, had a significant impact on OS (test of null hypothesis H0 *p* = 0.001) but not on DFS (test of null hypothesis H0, *p* = 0.117). Table 4 depicts the hazard ratio of each covariate and its impact on the overall model.

## 4. Discussion

Our study demonstrates that the PSS technique is not inferior to major resections in terms of oncological outcomes, with similar DFS and OS. Moreover, the PSS technique appears to be superior in terms of morbidity, major complications (CD III-IV), reintervention rate, and length of ICU stay. When performing PSM, the postoperative outcomes were equivalent except for morbidity where the results were still in favor of PSS. The shift in results after PSM must be taken with caution as for some variables the number of events was low, less than 3. Whenever technically possible, PSS should be the preferred approach. Even in cases of necessary R1 resections (parenchymal or vascular), outcomes are often comparable to R0 resections due to advancements in systemic chemotherapy [11,12]. As shown in our Cox multivariate survival analysis, positive resection margin (PRM) was not found to impact OS and DFS. More so, PRM was the least impactful covariate when compared to size or number of lesions. Decisions regarding the feasibility of PSS, particularly in patients with extensive disease (e.g., 10, 20, or 30 metastases), must be individualized and guided by multidisciplinary team (MDT) discussions, considering tumor biology, patient condition, and the expertise of the hepatobiliary surgeon. Major hepatectomy is generally indicated in the presence of multiple, large metastases involving three or more hepatic segments; invasion of major or sectoral intrahepatic portal pedicles; and infiltration of major hepatic veins not amenable to vascular reconstruction or inability to achieve R0 resection via PSS.

For a long time, MLR was favored for treating colorectal liver metastases due to its ability to provide a significant surgical margin [6,7,21]. However, since 2005, Pawlik et al. have demonstrated that a 1 mm resection margin provides the same overall survival (OS) as a wider margin [8]. Hamady et al., in a propensity score-matched analysis of nearly 3000 CRLM hepatectomies, also demonstrated that a 1 mm resection margin provides curative surgery, as extending the margin beyond 1 mm did not confer additional liver disease-free survival benefits [22]. Conversely, Kokudo et al. demonstrated that major hepatectomy does not necessarily lower intrahepatic recurrence rates. Their study reported an ipsilateral recurrence rate of less than 20% in patients who underwent nonanatomical PSS, compared to those who underwent major anatomical resection [23]. These findings have prompted many surgeons to adopt a more conservative surgical approach. While it is true that MLR generally achieves a higher R0 resection rate than PSS and is associated with a lower local recurrence rate, this does not have a statistically significant impact on overall survival. A 2023 meta-analysis by Wang et al., which included 7228 CRLM patients, found that PSS was linked to a slightly higher rate of repeat resections, an increased incidence of positive margins, and a greater risk of intrahepatic recurrence [15]. Despite these factors, long-term outcomes—such as OS, DFS, and liver DFS—showed no significant differences between the PSS and MLR groups. In our study, PSS seems to have a higher rate of margin positivity; however, the difference did not reach statistical significance (*p* = 0.487). More so, positive resection margin rate was included as a covariate in a multivariate Cox proportional hazards model of OS and DFS and was found to be the least impactful variable when compared to PSS, size of metastatic lesion, and number of lesions. In our practice, MLR was more commonly performed in the early years of the study. Currently, PSS has become the standard approach, with MLR reserved for carefully selected cases where PSS is not a viable option.

Critics of major liver resections for colorectal liver metastases acknowledge their potential for achieving oncological clearance but highlight several drawbacks, especially when compared to PSS. These include higher morbidity and mortality rates, an increased risk of liver insufficiency, potential overtreatment in certain cases, and greater surgical trauma leading to prolonged recovery [6,7,24,25]. A 2019 systematic review by Deng et al., which included 18 studies with a total of 7081 CRLM patients, demonstrated that PSS was associated with improved perioperative outcomes, including reduced postoperative complications, lower transfusion requirements, decreased blood loss, and shorter operating times. However, there were no significant differences in overall survival or recurrence-free survival between the PSS and MLR groups [17]. Wang et al., in a meta-analysis of 22 studies, found that PSS was associated with better perioperative outcomes, including shorter operating time, reduced intraoperative bleeding, lower blood transfusion rates, fewer postoperative complications, and a lower 90-day mortality rate compared to MLR [15]. In our study, MLR was associated with a higher risk of medical and surgical postoperative complications, including bleeding, surgical site infections, intra-abdominal collections, acute kidney injury, multiple organ dysfunction syndrome, and the need for reoperation despite having a significantly higher rate of males included, which could be regarded as a potential risk factor for morbidity. This increased risk was linked to longer operating times and extended ICU stays. Conversely, PSS was associated with a shorter hospital stay, with a mean duration of 12.3 days compared to 14.1 days for MLR (*p* = 0.109), contributing to a reduction in hospitalization-related costs (Table 2). Overall, there was a higher morbidity rate in the MLR group of 57.1% versus 11% for PSS patients (*p* = 0.001) and a significantly higher rate of major complications (Clavien–Dindo III–IV) (*p* = 0.005). Also, MLR was found to have a higher rate of reintervention (*p* = 0.023). In our cohort, out of the three reinterventions in the MLR group, only one was due to an intrinsic liver complication (i.e., bile leak) while the other two were performed due to postoperative small bowel obstruction (one for small bowel volvulus and one for small bowel intussusception). After performing PSM, the results favoring PSS downshifted; however, a significantly lower morbidity rate (*p* = 0.023) remained, emphasizing that even when matched on preoperative characteristics, PSS has a lower morbidity rate.

The presence of micrometastatic CRLMs in the future liver remnant has become a key argument against MRL [26]. Therefore, expanding the resection volume is a one-time strategy that may not benefit patients. The likelihood of reoperation in case of recurrence in the remnant liver is very low, limiting the patient’s opportunity for a new surgical line of treatment. Beyond maximizing future liver remnant (FLR) and reducing the risk of liver failure, perhaps the most significant advantage of PSS is enabling repeat hepatectomies, increasing the potential for future liver surgeries in cases of recurrence [27]. One recent meta-analysis emphasized that the greatest advantage of PSS lies in its ability to provide more treatment options following recurrence, particularly by increasing the likelihood of reoperation, ultimately contributing to prolonged survival [15]. Therefore, we believe that PSS should be regarded as a precision surgical strategy for detecting and managing metastases, much like systemic chemotherapy, by utilizing multiple treatment stages to address successive waves of recurrence. In our study, 3 patients with PSS underwent repeat hepatectomies while none did in the MLR group (5.2 vs. 0%).

The concept of PPS was pioneered in the 1980s by Makuuchi and colleagues, as a safer alternative to high-risk major resections [28]. Initially, PSS was often equated with limited resections for small, superficial CRLM—commonly known as “cherry-picking surgery”—or minor anatomic resections for deep-seated tumors (Figure 4). However, the advent of intraoperative ultrasound has transformed PPS by allowing surgeons to precisely navigate deep-seated tumors through multiplanar dissection. Intraoperative ultrasound enhances the ability to safely separate tumors from major intrahepatic vessels, provided there is no sign of vascular infiltration. Moreover, it enables visualization of communicating vessels between major hepatic veins, optimizing surgical planning. These advancements have paved the way for complex PPS techniques, including sophisticated core hepatectomies, thereby broadening the scope of parenchyma-sparing approaches in the management of CRLM [29,30,31]. In our study, all patients underwent intraoperative ultrasound-guided resections, ensuring precise navigation for deep metastases near major vessels, with complex resections consistently performed under IOUS guidance (Figure 5).

Currently, there is no universally accepted definition of PSS. Intuitively, it refers to a resection in which the minimal amount of peritumoral healthy tissue is excised to achieve an R0 resection. However, the challenge lies in defining what constitutes this “minimum” and determining a specific cut-off value. Evrard et al. proposed an innovative approach using a machine to slice the surgical specimen and analyze it with software, calculating the ratio of healthy tissue to tumor within the entire specimen [32]. Based on this method, a resection containing 66% healthy tissue and 34% tumor tissue could be classified as a PSS resection. While this may be applicable to small, localized tumors, the question remains whether the same threshold applies to larger or multiple tumors. Establishing a universal cut-off value is challenging and likely requires case-by-case consideration. A key debate among authors is whether a major resection that removes only minimal peritumoral healthy tissue should be classified as PSS or MLR. We argue that once a major resection is performed, regardless of the volume of healthy parenchyma removed, it carries the benefits and drawbacks of an MLR and should therefore be classified as such (Figure 6).

PSS may be particularly advantageous for patients undergoing systemic therapy prior to surgery, as neoadjuvant chemotherapy—especially oxaliplatin-based regimens—has been associated with sinusoidal obstruction syndrome (SOS), a chemotherapy-induced hepatic injury [33]. SOS has been linked to an increased risk of postoperative complications, such as bleeding, liver abscesses, and liver failure [34]. In our study, 69 out of 86 patients (80.2%) received neoadjuvant chemotherapy including oxaliplatin, with 44 patients (75.8%) in the PSS group and 25 patients (89.2%) in the MLR group (*p* = 0.246). Notably, in one patient in the MLR group who developed post-hepatectomy liver failure (PHLF) syndrome, the intraoperative evaluation revealed a characteristic post-chemotherapy “blue-liver” appearance (Figure 7).

Parenchymal-sparing resection can be effectively performed using laparoscopic techniques, further enhancing its benefits by minimizing invasiveness and promoting faster postoperative recovery [35]. There is a growing trend toward minimally invasive approaches for select liver metastases. The OSLO-COMET randomized controlled trial supported the use of laparoscopy for parenchymal-sparing liver resections, demonstrating fewer postoperative complications, shorter hospital stays, and comparable negative resection margins [36]. Moreover, a recent meta-analysis highlighted the advantages of laparoscopic PPS when combined with simultaneous colorectal resections [37].

There are limitations to our study. Firstly, the low number of patients in each group and the low number of events limit the power of the study. Secondly, the two groups were unbalanced (58 vs. 28), although propensity score matching was performed in a one-to-one fashion. Even though the data were retrieved by a surgical resident and confirmed by two senior surgeons, the study’s retrospective design suggests a significant risk of selection, recollection, misclassification bias, and potential data loss. Also, the small number of patients resulted in outcomes with few observations, often fewer than five, which reduced the statistical power of our analysis, especially after PSM when the number of patients in each group was reduced to ten. The study period spanned over ten years (2013–2024), which limits the validity of results considering the evolution of techniques over time and change in systemic treatment plans, although all surgeries were performed by a single surgeon who adhered to his technique for both MLR and PSS (when adopted, more recently). With regard to the survival curves, the log-rank test was limited in value, considering the crossing of hazard rates. For this reason, we have performed both Wilcoxon and Tarone–Ware tests, which are more sensitive to events at earlier time points, increasing the validity of our results. Despite its limitations, this study acknowledges the superiority of PSS in achieving better postoperative outcomes without compromising oncological outcomes, emphasizing its role as a standard surgical approach when possible. Future clinical trials should be encouraged to clearly define the role of PSS and better stratify its indication, considering covariates relating to the use of neoadjuvant therapy, possibility of obtaining negative margins, tumor biology, primary tumor location, and size/number of liver lesions.

## 5. Conclusions

Our study demonstrates that PSS provides oncologic outcomes and safety comparable to MLR in terms of OS and DFS. Additionally, PSS is associated with a lower morbidity rate. By preserving hepatic parenchyma and allowing for the possibility of future reoperations, PSS should be considered a viable surgical option for CRLM patients whenever technically feasible.

## Figures and Tables

**Figure 1 diagnostics-15-01334-f001:**
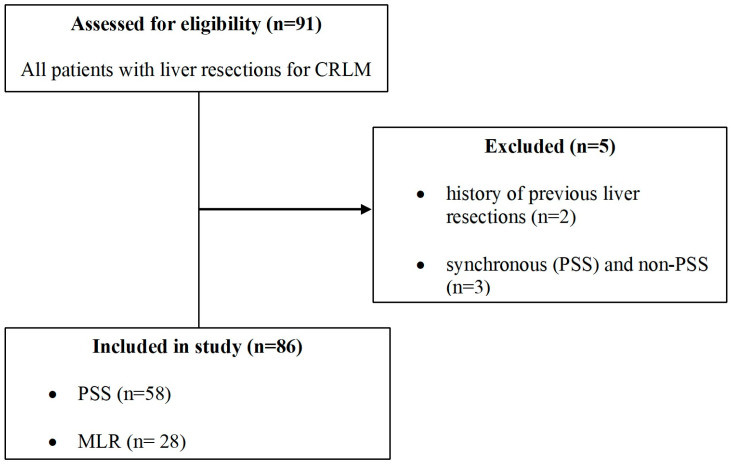
STROBE flowchart of eligibility.

**Figure 2 diagnostics-15-01334-f002:**
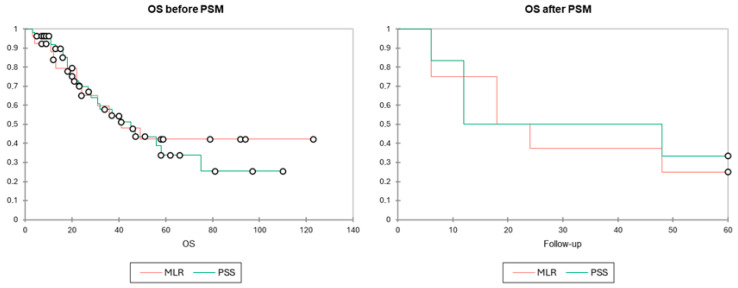
Kaplan–Meier OS distribution comparing major liver resections (red) vs. parenchymal sparing surgery (blue). Tests of equality of the survival distribution function (DF = 1) were conducted including log-rank test (*p* = 0.722), Wilcoxon test (*p* = 0.884), and Tarone–Ware (*p* = 0.951). All showed similar survival curves between the two groups before and after PSM.

**Figure 3 diagnostics-15-01334-f003:**
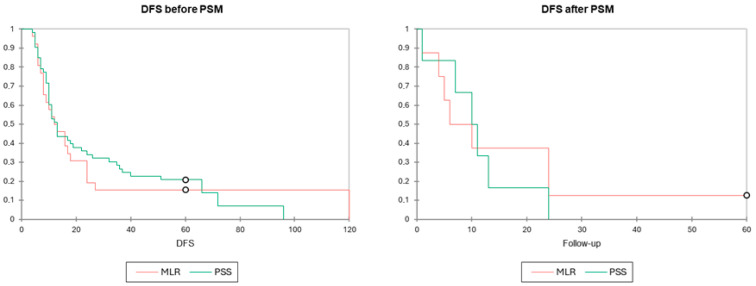
Kaplan–Meier DFS distribution comparing major liver resections (red) vs. parenchymal sparing surgery (blue). Tests of equality of the survival distribution function (DF = 1) were conducted including log-rank test (*p* = 0.667), Wilcoxon test (*p* = 0.519), and Tarone–Ware (*p* = 0.518). All showed similar survival curves between the two groups before and after PSM.

**Figure 4 diagnostics-15-01334-f004:**
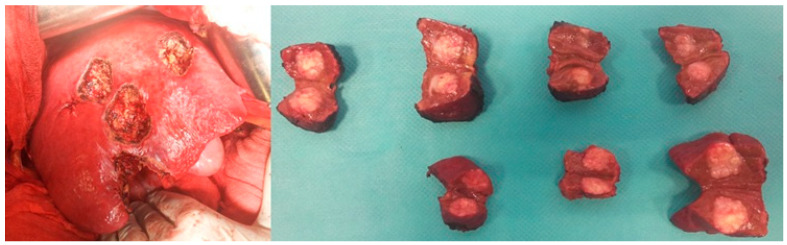
Minor PSS (“cherry-picking surgery”) for subcapsular lesions located in segments III, IV, V, VI, and VIII.

**Figure 5 diagnostics-15-01334-f005:**
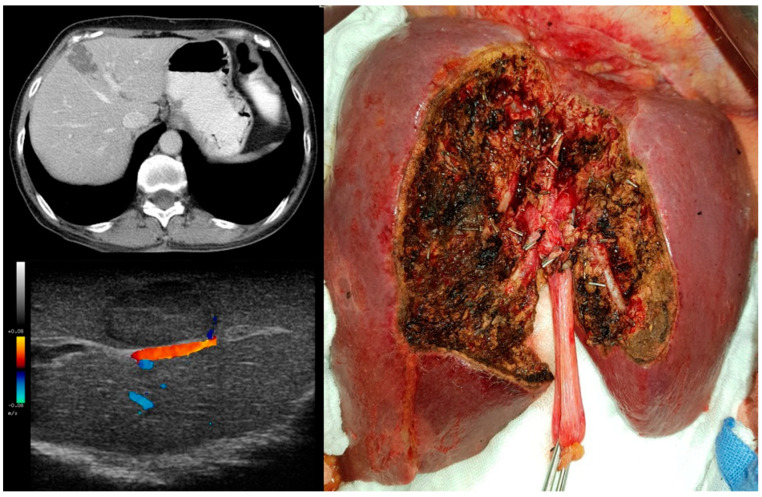
Patient with PSS for three liver metastases with intraoperative ultrasound-guided resection. The metastases were in close proximity to vascular structures but without vascular invasion, allowing for successful application of the PSS resection technique for all three lesions.

**Figure 6 diagnostics-15-01334-f006:**
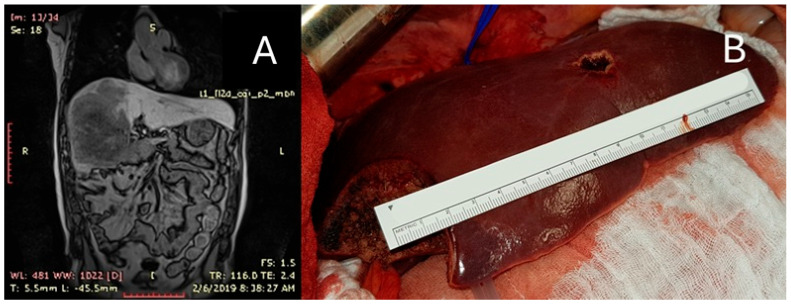
PSS as an extensive hepatic resection for CRLMs: Patient with right trisectionectomy for voluminous metastasis of rectosigmoid cancer; hepatic volumetry—tumor volume 2592 cm^3^, specimen volume 3030 cm3; resected non-tumor parenchyma volume only 14.6% of the specimen. For this patient, a small liver metastasis of 13 cm^3^ from segment III was also resected as PSS. (**A**) Preoperative MRI shows a large tumor occupying the entire right liver and almost completely segment IV. (**B**) Intraoperative image of remnant liver (segments II and III with PSS resection of segment III 13 cm3 metastasis) representing 62% of total non-tumor liver.

**Figure 7 diagnostics-15-01334-f007:**
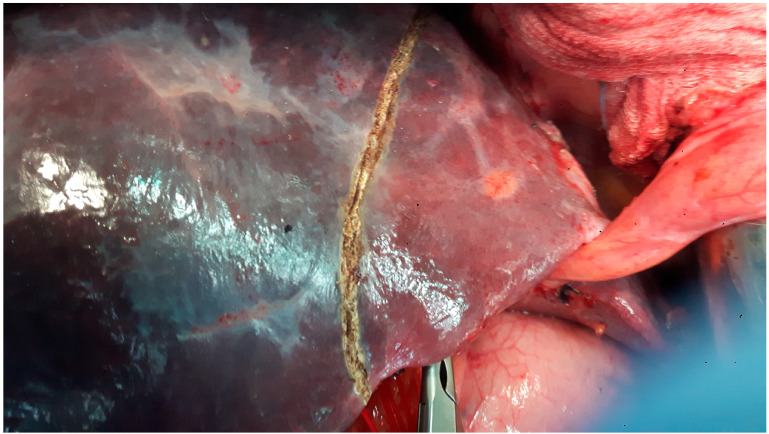
Patient with “blue liver” post-treatment with oxaliplatin, who required a right hepatectomy and who developed postoperative liver failure.

**Table 1 diagnostics-15-01334-t001:** Two-group comparison in terms of covariates and preoperative characteristics.

	PSS n (%)/Mean (SD)	MLR n (%)/Mean (SD)	*p*-Value
**Total**	**58 (100)**	**28 (100)**	
Gender (males)	39 (67.2)	12 (42.8)	***p* = 0.031**
Age, mean (SD)	60.7 (10.2)	59.8 (7.6)	*p* = 0.679
BMI	25.2 (3.8)	26 (3.3)	*p* = 0.343
mFI ≥ 2	6 (10.3)	7 (25)	*p* = 0.107
**Comorbidities**			
ALD	1 (1.7)	0	*p* = 1
Viral Hepatitis	0	2 (7.1)	*p* = 0.103
Cirrhosis	0	0	
Diabetes	0	0	
CHF	3 (5.1)	2 (7.1)	*p* = 0.658
COPD	2 (3.4)	1 (3.5)	*p* = 1
CKD	6 (10.3)	2 (7.1)	*p* = 1
Smoking	4 (6.8)	5 (17.8)	*p* = 0.143
Neoadjuvant therapy	44 (75.8)	25 (89.2)	*p* = 0.246
**ASA 2**	10 (17.2)	6 (21.4)	*p* = 0.768
**ASA 3**	43 (74.1)	20 (71.4)	*p* = 0.799
**Preoperative blood tests**			
Total Bilirubin (mg/dL)	0.70 (0.3)	0.76 (0.77)	*p* = 0.604
Albumin (g/dL)	4.54 (0.7)	4.6 (0.4)	*p* = 0.675
INR	1.00 (0.07)	1.03 (0.06)	*p* = 0.054
PLT (mmc) × 10^3^	227 (67)	258 (96)	*p* = 0.085
AST	29.9 (12.1)	33.7 (14.4)	*p* = 0.203
ALT	29.6 (21.7)	32.7 (19.8)	*p* = 0.525
**Operative factors**			
No. of liver lesions	1.6 (1.04)	1.67 (1.61)	*p* = 0.808
Operative time (min)	235.2 (77.9)	302.6 (127.3)	***p* = 0.003**
Blood loss (mL)	567.6 (422.2)	632.6 (533.8)	*p* = 0.541
Positive resection margin	8 (13.8)	2 (7.14)	*p* = 0.487

**Key:** Fisher exact test was used, instead of Chi-square test, to compare the number of events between groups as for some variables there were less than 5 observations. A two-sample *t*-test (pooled variance) was used to compare means. PSS, parenchymal-sparing surgery; MLR, major liver resection; SD, standard deviation; MD, mean difference; BMI, body mass index; mFI, modified frailty index; ALD, alcoholic liver disease; CHF, chronic heart failure; COPD, chronic obstructive pulmonary disease; CKD, chronic kidney disease; RT, radiotherapy; CHT, chemotherapy; RCHT, radio-chemotherapy; ASA, American Society of Anaesthesiologists Classification; INR, International Normalized Ratio; PLT, platelets; AST, Aspartate Transferase; ALT, Alanine Transaminase.

**Table 2 diagnostics-15-01334-t002:** Postoperative outcomes in PSS and MLR before and after PSM.

	Before PSM			After PSM		
	PSS n (%)/Mean (SD)	MLR n (%)/Mean (SD)	*p*-Value	PSS n (%)/Mean (SD)	MLR n (%)/Mean (SD)	*p*-Value
**Total**	**58 (100)**	**28 (100)**		**10 (100)**	**10 (100)**	
Morbidity	11 (18.9)	16 (57.1)	***p* = 0.001**	2 (20)	8 (80)	***p* = 0.023**
Clavien–Dindo III–V	0	5 (17.8)	***p* = 0.005**	0	2 (20)	*p* = 0.473
**Surgical complications**						
Bile leak	1 (1.7)	2 (7.1)	*p* = 0.246	0	0	
PHLF	0	2		0	0	
Bleeding/Hematoma	0	2 (7.1)	*p* = 0.103	0	0	
IAC	1 (1.7)	2 (7.1)	*p* = 0.246	0	0	
SSI	0	1 (3.5)	*p* = 0.325	0	1 (10)	*p* = 1
HAI	4 (6.8)	3 (10.7)	*p* = 0.690	1 (10)	3 (30)	*p* = 0.582
**Medical complications**						
DVT/PE	0	0		0	0	0
HAP	3 (5.1)	1 (3.5)	*p* = 1	1 (10)	1 (10)	*p* = 1
Cardiac	0	1 (3.5)	*p* = 0.325	0	1 (10)	*p* = 1
AKI	2 (3.4)	3 (10.7)	*p* = 0.324	0	1 (10)	*p* = 1
MODS	0	1 (3.5)	*p* = 0.325	0	1 (10)	*p* = 1
Reintervention	0	3 (10.7)	***p* = 0.032**	0	2 (20)	*p* = 0.473
LOS	12.3 (4.1)	14.1 (6.1)	*p* = 0.109	14.7 (6.2)	13.3 (4.8)	*p* = 0.579
Length of ICU stay	2.1 (1.7)	3.5 (4.7)	***p* = 0.046**	2.6 (2.2)	5 (7.7)	*p* = 0.355
30-day mortality	0	0		0	0	

**Key:** Fisher exact test was used, instead of Chi-square test, to compare the number of events between groups as for some variables there were less than 5 observations. Two-sample *t*-test (pooled variance) was used to compare means. PSM, propensity score matching; PSS, parenchymal-sparing surgery; MLR, major liver resection; SD, standard deviation; MD, mean difference; PHLF, post hepatectomy liver failure; IAC, intraabdominal collection; SSI, surgical site infection; HAI, hospital-acquired infection; DVT, deep vein thrombosis; PE, pulmonary embolism; HAP, hospital-acquired pneumonia; AKI, acute kidney injury; MODS, multiple organ dysfunction syndrome, LOS, length of stay.

**Table 3 diagnostics-15-01334-t003:** OS and DFS comparison between PSS and MLR before and after PSM.

	Before PSM				After PSM			
	**OS (months)**		**DFS (months)**		**OS (months)**		**DFS (months)**	
	Median (Range)	Wilcoxon text	Median (Range)	Wilcoxon text	Median (Range)	Wilcoxon text	Median (Range)	Wilcoxon text
**PSS**	31 (3–110)	*p* = 0.884	25.2 (4–96)	*p* = 0.519	40.9 (8–97)	*p* = 0.741	24.3 (7–72)	*p* = 0.653
**MLR**	37.1 (3–123)		22.2 (4–120)		21.2 (3–59)		13.8 (6–24)	

**Key:** PSM, propensity score matching; OS, overall survival; DFS, disease-free survival; PSS, parenchymal-sparing surgery; MLR, major liver resection; SD, standard deviation.

**Table 4 diagnostics-15-01334-t004:** Cox proportional hazards survival at mean of covariates (OS and DFS).

OS			
**Variable**	**Wald Chi^2^**	***p*-value**	**Hazard ratio**
**PSS**	3.662	0.056	2.177 [0.981–4.827]
**PRM**	0.732	0.392	0.608 [0.195–1.899]
**Size**	20.727	<0.0001	6.895 [3.003–15.832]
**Number**	1.398	0.237	1.672 [0.713–3.918]
**DFS**			
**Variable**	**Wald Chi^2^**	***p*-value**	**Hazard ratio**
**PSS**	0.007	0.935	0.978 [0.570–1.676]
**PRM**	0.154	0.695	0.855 [0.392–1.866]
**Size**	1.254	0.263	1.395 [0.779–2.497]
**Number**	7.763	0.005	2.451 [1.305–4.606]

**Key:** OS, overall survival; DFS, disease-free survival; PSS, parenchymal-sparing surgery; PRM, positive resection margins; Size, greatest diameter of metastasis more than 50 mm; Number, more than two lesions.

## Data Availability

The data that support the findings of this study are available on request from the corresponding author, S.M.

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
