# Peer review of "Can We Achieve More with Less? Parenchymal Sparing Surgery Versus Major Liver Resection for Colorectal Liver Metastases: An Observational Single-Center Study with Propensity Score Analysis"

_diagnostics, 2025, doi:10.3390/diagnostics15111334_

Round 1
Reviewer 1 Report
Comments and Suggestions for Authors
- The sample size of the PSS group (n=58) and the MLR group (n=28) in this study was unbalanced. This imbalance may affect the statistical power of the study and the reliability of the comparison.
- The study notes that positive resection margins were more common in the PSS group (13.8% vs. 7.14%), but it concludes that this had no impact on survival. Without further analysis or discussion of potential confounding factors, this conclusion may be premature.
- For the outcome (mean survival time), the mean is more susceptible to extreme values, while the median is more resistant to outliers.
- Some log-rank curves are crossed over, and the log-rank test may not be applicable.
- The t-test for quantitative variables was mentioned in the text, but the normality assumption was not discussed.
- The software used for statistical analysis (other than XLSTAT) is not mentioned in the text.
- The text points out that the number of male patients in the PSS group was significantly higher (39 vs. 12, p=0.03). While this is statistically significant, the clinical significance of this difference is not discussed.
- This study is a propensity score analysis. However, the results presented in the Abstract are data before PSM (propensity score matching). The only difference (P=0.023) between PSS and MLR after PSM is morbidity.
- Spelling errors should be corrected, e.g., Wilcoxon text.
Author Response
Manuscript ID: diagnostics-3586529
Title: Can we achieve more with less? Parenchymal Sparing Surgery versus Major Liver Resection for Colorectal Liver Metastases: an observational single-centre study with propensity score analysis
Response to reviewers
Dear Editor,
Thank you for giving us the opportunity to submit a revised manuscript in your esteemed journal. We appreciate the time and effort that you and the reviewers dedicated to our study, and we are grateful for the interesting comments and valuable suggestions you have made to the paper.
In this revised manuscript we have incorporated most of the comments made by the reviewers and highlighted the changes to the text in red. Please find bellow our point-by-point replies and changes to the reviewers’ suggestions.
REVIEWER 1
Comment 1: The sample size of the PSS group (n=58) and the MLR group (n=28) in this study was unbalanced. This imbalance may affect the statistical power of the study and the reliability of the comparison.
Answer: Thank you for this. Indeed the unequal group size reduces the power of our study, and in the revised version we emphasized this, in the Discussion section were we presented our study limitations, however we did perform PSM which calibrates groups in a one to one fashion.
Comment 2: The study notes that positive resection margins were more common in the PSS group (13.8% vs. 7.14%), but it concludes that this had no impact on survival. Without further analysis or discussion of potential confounding factors, this conclusion may be premature.
Answer: Thank you for this remark. Indeed, positive resection margin seems to be more common in the PSS group, but the difference is not statistically significant (p=0.487), thus we cannot draw any conclusions. Even so, as other previously published papers reported, the higher rate of margin positivity does not seem to impact DFS thanks modern systemic therapy. Even in the case of R1 resection the addition of adjuvant therapy can improve outcomes to match those of R0 resections. We did emphasize this in the revised version in Discussions.
Comment 3: For the outcome (mean survival time), the mean is more susceptible to extreme values, while the median is more resistant to outliers.
Answer: Thank you for this remark. Indeed, median survival time is insensitive to outliers so we changed all means to median and range. The overall significance did not change.
Comment 4: Some log-rank curves are crossed over, and the log-rank test may not be applicable.
Answer: Thank you for pointing this out. Given the presence of crossing rates, the log-rank test does lose its power becoming inadequate. For this reason, we also performed Wilcoxon test which gives more weight to events at early times points compared to log-rank which is more sensitive to distributional differences late in time. Similarly, the Tarone-Ware test could be more appropriate in crossing curves and we used it also. Regardless, the results have limitations given the low number of patients and events. In the revised version we have explained this at study limitations and described more the Wilcoxon test results, rather than log-rank.
Comment 5: The t-test for quantitative variables was mentioned in the text, but the normality assumption was not discussed.
Answer: Thank you for your observation. We agree that the assumption of normality is essential when applying parametric tests such as the t-test. We have now added a clarification in the Methods section indicating that normality was assessed using the Shapiro-Wilk test and that the data met the assumption for normality.
Comment 6: The software used for statistical analysis (other than XLSTAT) is not mentioned in the text.
Answer: Thank you. We used XLSTAT (Lumivero®) to perform all our data analysis including survival analysis, not only PSM, as the initial text was suggesting. We have amended Materials and Methods clarifying this.
Comment 7: The text points out that the number of male patients in the PSS group was significantly higher (39 vs. 12, p=0.03). While this is statistically significant, the clinical significance of this difference is not discussed.
Answer: Thank you for pointing this out. We have added a new paragraph in the discussions stating the significance of this finding. Indeed, more men could be potentially regarded as a risk factor for morbidity. Despite having more males, PSS still have better postoperative outcomes, lower morbidity and a lower rate of major (CD III-IV) complications.
Comment 8: This study is a propensity score analysis. However, the results presented in the Abstract are data before PSM (propensity score matching). The only difference (P=0.023) between PSS and MLR after PSM is morbidity.
Answer: Thank you for your observation. In the revised version we have added the PSM data also, both for postoperative outcomes and for survival. However, we also kept the data before PSM as we believe it to still be valid for some variables where the number of events was very low (less than 3) after PSM.
Comment 9: Spelling errors should be corrected, e.g., Wilcoxon text.
Answer: We reviewed the text and amended all typos.
Reviewer 2 Report
Comments and Suggestions for Authors
INTRODUCTION: The author appropriately references the existing literature supporting the effectiveness of PSS. However, a brief and explicit discussion of the methodological limitations commonly found in these studies should be included, such as the predominant use of retrospective designs, population heterogeneity, and the scarcity of data from randomized controlled trials. Furthermore, it would be advisable to clearly anticipate how certain confounding variables, including the biological characteristics of the metastases, the primary tumor location, and the use of preoperative systemic treatments (e.g., neoadjuvant chemotherapy), may influence oncologic outcomes and, consequently, the surgical choice between PSS and major liver resection.
MATERIALS AND METHODS: Given the retrospective design of the study, it is essential that the authors clearly acknowledge the intrinsic limitations of this methodology, such as selection bias (due to non-random patient assignment), misclassification bias, and potential data loss during follow up. The specific clinical criteria that determined patient allocation to the PSS or MLR group should be detailed to clarify the potential impact of initial selection bias. Additionally, the authors should briefly explain the rationale for choosing such a long study period (2013–2024) and analyze whether changes in surgical techniques or outcomes occurred over time. Finally, the follow-up protocols must be described more clearly, including the frequency of clinical visits and imaging assessments, as well as the standard duration of follow-up.
RESULTS: The findings indicate that specific complications were comparable between the two surgical groups; however, the MLR group experienced significantly more reinterventions and longer ICU stays. The authors should clearly specify the clinical reasons and particular complications that led to these reinterventions in the MLR group. Moreover, after propensity score matching, the sample size was substantially reduced (n=10 per group), which significantly limits the statistical power to draw definitive conclusions. This methodological limitation must be explicitly discussed to prevent over interpretation of the results. It is also advisable to include a summary table reporting a Cox multivariate analysis incorporating known prognostic factors, such as surgical margins, number and size of metastases, and use of neoadjuvant chemotherapy, to better elucidate the impact of these variables on survival outcomes.
DISCUSSION and CONCLUSIONS: The discussion should be expanded to better define the specific clinical indications that still justify the use of major liver resection, clarifying in which clinical scenarios this approach remains preferable over parenchymal-sparing surgery. Additionally, the authors should more clearly define the clinical and technical criteria that determine when PSS is technically feasible, thereby offering practical guidance for surgical decision-making. Finally, the main methodological limitations of the study (retrospective nature, small sample size after matching) should be explicitly mentioned, and the need for future prospective or randomized trials should be emphasized to more robustly confirm the findings presented.
Author Response
Dear Reviewer,
Thank you for reviewing our paper. We appreciate the time and effort that you dedicated to our study, and we are grateful for the interesting comments and valuable suggestions you have made to the paper.
In this revised manuscript we have incorporated most of the comments made by you and highlighted the changes to the text in red. Please find bellow our point-by-point replies and changes.
REVIEWER 2
Comment 1: The author appropriately references the existing literature supporting the effectiveness of PSS. However, a brief and explicit discussion of the methodological limitations commonly found in these studies should be included, such as the predominant use of retrospective designs, population heterogeneity, and the scarcity of data from randomized controlled trials. Furthermore, it would be advisable to clearly anticipate how certain confounding variables, including the biological characteristics of the metastases, the primary tumor location, and the use of preoperative systemic treatments (e.g., neoadjuvant chemotherapy), may influence oncologic outcomes and, consequently, the surgical choice between PSS and major liver resection.
Answer: Thank you for your suggestions. We have added a new paragraph in Introduction discussing these perspectives [Lines 60-71].
Comment 2: Given the retrospective design of the study, it is essential that the authors clearly acknowledge the intrinsic limitations of this methodology, such as selection bias (due to non-random patient assignment), misclassification bias, and potential data loss during follow up. The specific clinical criteria that determined patient allocation to the PSS or MLR group should be detailed to clarify the potential impact of initial selection bias. Additionally, the authors should briefly explain the rationale for choosing such a long study period (2013–2024) and analyze whether changes in surgical techniques or outcomes occurred over time. Finally, the follow-up protocols must be described more clearly, including the frequency of clinical visits and imaging assessments, as well as the standard duration of follow-up.
Answer: Thank you for your observations. The retrospective design suggests a significant risk of selection, recollection, and misclassification bias. Also, the small number of patients resulted in outcomes with few observations, often fewer than five, which reduced the statistical power of our analysis, especially after PSM when to number of patients in each group was reduced to ten. The study period spanned over ten years (2013-2024) which limits validity of results considering evolution of technique over time and change in systemic treatment plans, although all surgeries were performed by a single surgeon which adhered to his technique for both MLR and PSS (when adopted, more recently). In the revised version we have explained (Discussions – study limitations, Lines 352-370)
Comment 3: The findings indicate that specific complications were comparable between the two surgical groups; however, the MLR group experienced significantly more reinterventions and longer ICU stays. The authors should clearly specify the clinical reasons and particular complications that led to these reinterventions in the MLR group. Moreover, after propensity score matching, the sample size was substantially reduced (n=10 per group), which significantly limits the statistical power to draw definitive conclusions. This methodological limitation must be explicitly discussed to prevent over interpretation of the results. It is also advisable to include a summary table reporting a Cox multivariate analysis incorporating known prognostic factors, such as surgical margins, number and size of metastases, and use of neoadjuvant chemotherapy, to better elucidate the impact of these variables on survival outcomes.
Answer: Thank you for your observations.
- Indeed, the MLR group had more reinterventions than the PSS which had none. In the revised version we have explained the indication for reintervention for each case (n=3). In two cases the reason for reintervention small bowel obstruction while in one case we had an MRI confirmed bile leak (Lines 278-285).
- After performing PSM the number of cases reduced to 10 vs 10 in one-to-one matching. Although it reduces selection and confounding bias, the low number of patients limits the power of the analysis, and we have highlighted this in the revised version at study limitations.
- In the revised version we have also added a new paragraph and Table in the Results section where we have performed, as suggested, a multivariate Cox proportional hazards analysis on OS and DFS including covariates such as PSS, positive resection margin rate, size of lesion and number of lesions. The use of neoadjuvant therapy could not be included in the model due to the low number of non-events over time follow-up timframe. Table 4 and Lines 207-217 have addressed your suggestion.
Comment 4: The discussion should be expanded to better define the specific clinical indications that still justify the use of major liver resection, clarifying in which clinical scenarios this approach remains preferable over parenchymal-sparing surgery. Additionally, the authors should more clearly define the clinical and technical criteria that determine when PSS is technically feasible, thereby offering practical guidance for surgical decision-making. Finally, the main methodological limitations of the study (retrospective nature, small sample size after matching) should be explicitly mentioned, and the need for future prospective or randomized trials should be emphasized to more robustly confirm the findings presented.
Answer: Thank you for your input.
- In the Discussion section we have added a new paragraph delineating the indications for PSS and when MLR could be considered as a better alternative. From our experience and according to literature, PSS should be the preferred approach whenever technically feasible as it is associated with similar oncological outcomes and better morbidity and short-term mortality. Decisions regarding the feasibility of PSS, particularly in patients with extensive dis-ease (e.g., 10, 20, or 30 metastases), must be individualized and guided by MDT discussions, considering tumour biology, patient condition, and the expertise of the hepatobiliary surgeon. Major hepatectomy is generally indicated in the presence of multiple, large metastases involving three or more hepatic segments; invasion of major or sectoral intrahepatic portal pedicles; infiltration of major hepatic veins not amenable to vascular reconstruction or inability to achieve R0 resection via PSS.
- Thank you. All the study limitations were added in the last paragraph of the Discussion section (Lines 381-399). Indeed, the retrospective nature, low number of patients, especially after PSM reduces the power of the analysis and imposes a high risk of selection, misclassification, confounding and loss of data bias. We have emphasized the need for future clinical trials that could clearly define the role of PSS and better stratify its indication considering covariates relating to use of neoadjuvant therapy, possibility of obtaining negative margins, tumour biology, primary tumour location and size/number of liver lesions.
Hope you will agree with our changes to the manuscript making it suitable for publication.
Kind regards,
The authors.
Round 2
Reviewer 1 Report
Comments and Suggestions for Authors
The authors have adequately addressed my comments in the revised version of the manuscript, so I have no further comments.